# Filamentation of the bacterial bi-functional alcohol/aldehyde dehydrogenase AdhE is essential for substrate channeling and enzymatic regulation

Pauline Pony [1,2], Chiara Rapisarda[1,2,4], Laurent Terradot [3], Esther Marza [1,2✉] & Rémi Fronzes [1,2✉]

Acetaldehyde–alcohol dehydrogenase (AdhE) enzymes are a key metabolic enzyme in bacterial physiology and pathogenicity. They convert acetyl-CoA to ethanol via an acetaldehyde intermediate during ethanol fermentation in an anaerobic environment. This two-step reaction is associated to $NAD^+$ regeneration, essential for glycolysis. The bifunctional AdhE enzyme is conserved in all bacterial kingdoms but also in more phylogenetically distant microorganisms such as green microalgae. It is found as an oligomeric form called spirosomes, for which the function remains elusive. Here, we use cryo-electron microscopy to obtain structures of *Escherichia coli* spirosomes in different conformational states. We show that spirosomes contain active AdhE monomers, and that AdhE filamentation is essential for its activity in vitro and function in vivo. The detailed analysis of these structures provides insight showing that AdhE filamentation is essential for substrate channeling within the filament and for the regulation of enzyme activity.

[1] Structure and Function of Bacterial Nanomachines – Institut Européen de Chimie et Biologie, University of Bordeaux, 2 rue Robert Escarpit, 33600 Pessac, France. [2] Microbiologie fondamentale et pathogénicité, UMR 5234, CNRS, University of Bordeaux, 2 rue Robert Escarpit, 33600 Pessac, France. [3] UMR 5086 Molecular Microbiology and Structural Biochemistry, Institut de Biologie et Chimie des Protéines, CNRS-Université de Lyon, Lyon, France. [4] Present address: UCB Celltech, Slough SL1 3WE, UK. ✉email: esther.marza@u-bordeaux.fr; remi.fronzes@u-bordeaux.fr

Facultative anaerobe bacteria, such as *Escherichia coli*, are metabolically versatile and are able to grow under a wide range of oxygen concentrations, from anaerobic conditions in the gut to aerobic environments. This metabolic adaptability is vital for the fitness and survival of commensal and pathogenic bacteria. In the absence or low concentrations of oxygen, bacteria use fermentation to survive. The bifunctional alcohol–acetaldehyde dehydrogenase enzyme (AdhE) is a key metabolic enzyme of the alcoholic fermentation pathway. This 96 kDa enzyme is highly expressed in the absence of oxygen and is essential for ethanol production in *E. coli*[1]. It is composed of a N-terminal aldehyde dehydrogenase (AlDH) domain and a C-terminal iron-dependent alcohol dehydrogenase (ADH) domain. AdhE converts acetyl-coenzyme A to acetaldehyde and then to ethanol, in a two-step reaction that is coupled with the oxidation of two NADH molecules into $NAD^+$. The biochemistry of AdhE enzymatic activities is well characterized[2,3]. In particular, a lot of investigations looked into this key fermentative enzyme to engineer novel ethanol-producing bacteria[4,5] or identify bacterial strains capable of growing under ethanol[6].

Interestingly, the biological role of AdhE seems to go beyond alcoholic fermentation. This protein could also be directly or indirectly involved in bacterial pathogenicity. In *Listeria monocytogenes*, the *Listeria* adhesion protein LAP (homologous to AdhE) promotes bacterial adhesion by interacting with a receptor of intestine cells[7,8]. It has also been reported in *Streptococcus pneumoniae* that AdhE also promotes bacterial adherence[9]. In *E. coli*, this enzyme would play a role during colonization by regulating the expression of key virulence genes[10]. Finally, AdhE could also act as a peroxide scavenger under aerobic conditions[11].

Although alcohol and aldehyde dehydrogenases are found as monofunctional enzymes in all kingdom of life, the bifunctional enzyme AdhE is found mostly in bacteria and in some unicellular eukaryotes such as microalgae. Intriguingly, AdhE is capable of oligomerization and forms a filament (identified as spirosome in early studies). These helical macromolecular assemblies were observed for the first time in the 1970s in bacteria[12]. These filaments of AdhE are widely conserved in bacteria[13] and also in phylogenetically distant organisms such as *Chlamydomonas reinhardtii*[2] and *Entamoeba histolytica*[14]. Although their wide conservation probably highlights an essential function, the exact role of AdhE filaments remains unknown. Recently, the cryo-electron microscopy (cryoEM) structure of the spirosome from *E. coli* was reported[15]. This study confirmed that AdhE is composed of canonical aldehyde dehydrogenase (AlDH) and ADH domains, interconnected by a short linker. As proposed in earlier studies[15,16], the AdhE oligomerization interfaces are mediated by canonical ADH–ADH and AlDH–AlDH dimerization interfaces. In the spirosomes, AdhE monomers are interlocked in a head-to-head manner through the AlDH–AlDH dimerization interface. These dimers are further assembled into a helical filament through ADH domain dimerization. Finally, it was shown that spirosomes display AlDH and ADH activities, and that the integrity of the ADH–ADH interface, which is essential for the spirosome assembly, is also essential for AdhE activity in vitro[15].

Here we used cryoEM to obtain structures of *E. coli* spirosomes in different conformational states using helical reconstruction (HR) and single-particle analysis (SPA). The detailed analysis of these structures provides insight showing that AdhE filamentation is essential for substrate channeling between the AlDH and ADH domains, and for the regulation of enzyme activity. Finally, we confirm that spirosomes contain active AdhE monomers and show that AdhE filamentation is essential for AlDH activity in vitro and AdhE function in vivo.

## Results

**CryoEM structure of *E. coli* spirosomes in different states.** In early studies, Kessler et al.[17] reported that the *E. coli* spirosomes were found in closed (compact) or open (extended) conformations and would change their conformation upon the addition of ligands. Strikingly, we and others reported the observation of native spirosomes isolated from various bacterial and eukaryotic species, all found exclusively in the open/extended conformation. The conformational change observed, which seems to depend on the binding of the cofactors, could be an important feature for the regulation of its activity.

The *adhE*[E. coli] gene was cloned in a high copy number vector and expressed in *BL21 E. coli* cells. Recombinant N-terminally $His_6$-tagged AdhE was purified by Nickel-nitrilotriacid affinity chromatography followed by gel filtration. The fraction containing filaments was collected in the void volume of the gel filtration column (Supplementary Fig. 1b) (see Methods section for details). Purified spirosomes were incubated with various combinations of ligands to determine the conditions, which trigger this conformational change (Fig. 1a). Samples were deposited on cryoEM grids and vitrified in liquid ethane. Micrographs of frozen-hydrated spirosomes were collected using a Talos Arctica cryo-electron miscoscope equipped with K2 summit direct electron detector. Contrast transfer function (CTF)-corrected and re-aligned movies were analyzed using RELION 3 software[18]. Non-overlapping helical segments were sorted with several rounds of two-dimensional (2D) classification. In the classes obtained, we could clearly distinguish the compact and extended conformations of the spirosomes, and we confirmed that in the absence of ligand (apo), the spirosomes are compact (Fig. 1a). Incubation with $NAD^+$ and $Fe^{2+}$ is sufficient to extend the filaments. The addition of Coenzyme A does not impair the conformational change triggered by $NAD^+$ and $Fe^{2+}$ (Fig. 1a). Interestingly, we show that in the same conditions, NADH and $Fe^{2+}$ are not able to trigger a conformational change from the compact to the extended form (Fig. 1a). However, it was reported that NADH could trigger the spirosomes extension[15]. Although similar conditions were used for protein and ligand concentrations, we could not reproduce this result.

The recently reported cryoEM structure of the *E. coli* spirosome was obtained in the absence of ligand (apo-form). Therefore, the spirosomes were observed in the closed/compact conformation. To obtain the structure of the *E. coli* spirosomes in their extended and compact form, both in the presence of ligands, we collected larger datasets of the spirosomes incubated with $NAD^+$, $Fe^{2+}$, and CoA or with NADH and $Fe^{2+}$. As in both cases the filaments were rather short and flexible, we chose to use SPA to solve their structure. However, we also used HR in parallel to uncover the helical symmetry parameters of these filaments and obtain a cryoEM map of the complete assemblies. For SPA, after 2D classification, an initial model was generated and refined in RELION. The structure corresponding to the AlDH and ADH domains were identified in the map. The part of the map corresponding to one AdhE dimer in interaction with two adjacent ADH domains (colored in Fig. 1b, c) were further refined by focused refinement. The final map was obtained after Bayesian polishing, subtraction, and focused refinement. A resolution of 3.2 Å and 3.8 Å was obtained for the extended and compact spirosome, respectively (Supplementary Fig. 2). For HR, after 2D classification, an initial map was reconstructed without any symmetry imposed using a feature-less cylinder as a three-dimensional (3D) reference. The helical symmetry of the AdhE filaments was determined in real space and was imposed during further refinements. The final 3D reconstruction was obtained after Bayesian polishing. The overall resolution of the

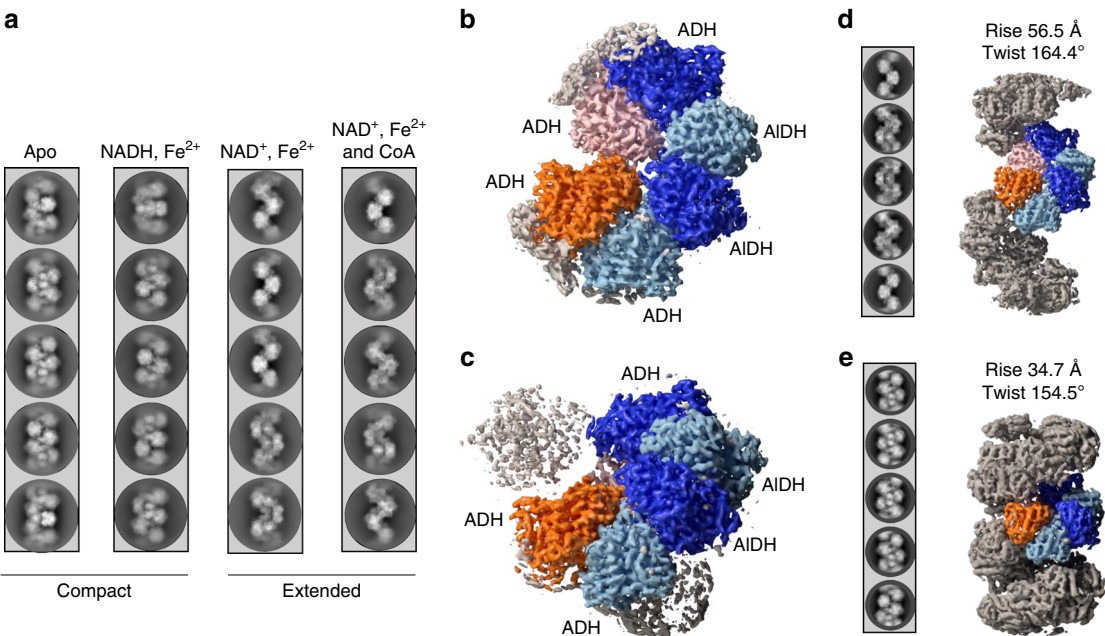

**Fig. 1 CryoEM analysis of the AdhE spirosomes in their compact and extended forms. a** Spirosomes were incubated with different cofactors as indicated in the panel. The condition *apo* corresponds to the spirosomes as they were purified from *E. coli*. For each condition, five representative 2D classes are displayed. **b** CryoEM map of the extended spirosomes (incubated with NAD$^+$, Fe$^{2+}$, and CoA) obtained by single-particle analysis. An AdhE dimer is colored with each AdhE protomer in blue and light blue, respectively. Adjacent ADH domains are colored in orange and pink. **c** CryoEM map of the compact spirosomes (incubated with NADH and Fe$^{2+}$) obtained by single-particle analysis. The same color codes as in **b** are used. **d** CryoEM map of the extended spirosomes obtained by helical reconstruction. Representative 2D classes of the helical segments are displayed and the refined helical parameters are indicated. The same color codes as in **b** are used. **e** CryoEM map of the compact spirosomes obtained by helical reconstruction. Representative 2D classes of the helical segments are displayed and the refined helical parameters are indicated. The same color code as in **b** are used.

final masked map was estimated to be 4.4 Å and 5 Å for the extended and compact spirosomes, respectively (Supplementary Fig. 2). The refined helical symmetry was 56.5 Å rise and 164.4° twist of the extended spirosome, and 34.7 Å and 154.5° twist for the compact spirosome. The maps used to reconstitute the complete spirosome assembly were obtained after imposing these symmetry parameters to the central 30% of the maps obtained (Fig. 1d, e and Supplementary Fig. 3).

**Structure of the AdhE filament in its extended state**. As this work was started before the publication of the work by Kim et al.[15], homology models of AdhE AlDH and ADH domains obtained using Swissmodel[19] were docked in the map of the extended spirosome obtained by SPA. A complete structural model was rebuilt inside the densities using Coot and then refined and validated using PHENIX[20,21] (see Methods section for details). As described previously[15], the AdhE monomer is composed of an N-terminal AlDH domain connected by a linker region to the C-terminal ADH domain (Fig. 2a). Each domain resembles the canonical AlDH and ADH domains found in the corresponding monofunctional enzymes. From N to C terminus of each domain, they are composed of a NAD-binding domain and catalytic domain. Two AdhE protomers are interlocked with apparent C2 symmetry to form the AdhE dimer (Fig. 2b). These protomers will be named (α or β) and their respective domains α-AlDH, α-ADH, β-AlDH, or β-ADH. Clear density for NAD$^+$ and Fe$^{2+}$ could be observed in the density map in the AlDH and ADH domains in both subunits of the AdhE dimer (Fig. 2a). The AdhE helical assembly can be described as one AdhE dimer repeated along a right-handed helix with a helical twist of 164.5 Å and a rise of 56.4 Å (Fig. 1c). To obtain a model of the AdhE filament, the AdhE dimer was docked in the map obtained by HR and duplicated using the helical parameters. Based on these

helical parameters and polarity, the AdhE dimers will be numbered $n$, $n + 1$, $n + \ldots$

**Structure of the AdhE filament in its compact state**. The structure of the AdhE spirosome in its compact state was obtained using the same strategy. The overall structure of the AdhE monomer is unchanged (Fig. 2d). Clear density for Fe$^{2+}$ and NADH could be found in the ADH domain (Fig. 2d). No density is visible for NADH within the AlDH domain. Within the AdhE dimer, both subunits are related by apparent C2 symmetry (Fig. 2e). Using the helical parameters and map obtained by HR, we could build a structural model for the spirosome in its compact state (Fig. 2f). We will use the same nomenclature for AdhE domains and protomers than described above. We notice that this structure is highly similar to the structure recently published in its apo-form[15] (Supplementary Fig. 4).

**Inter-domain interactions within the extended AdhE filament**. In the extended AdhE filament, three different inter-domain interfaces are formed. Within each AdhE dimer, two interfaces are found. The dimer is maintained through an AlDH–AlDH interface (named (α-AlDH)$n$/(β-AlDH)$n$ interface) and two symmetric AlDH–ADH interfaces (named (α-AlDH)$n$/(β-ADH)$n$ and (α-ADH)$n$/(β-AlDH)$n$ interfaces) (Fig. 2b, e and Supplementary Fig. 5). Along the helical assembly, on both sides of the dimer $n$, are found two equivalent interfaces between ADH domains (named (α-ADH)$n − 1$/(β-ADH)$n$) and (α-ADH)$n$/(β-ADH)$n + 1$). The (α-AlDH)$n$/(β-AlDH)$n$ interface is characterized by hydrogen bonds (Supplementary Figs. 5a and 6b). The (α−AlDH)$n$/(β-ADH)$n$ and (α-ADH)$n$/(β-AlDH)$n$ interfaces are maintained by five salt bridges and a vast network hydrogen bonds scattered along the interfaces (Supplementary Figs. 5b and 6a). Moreover, the dimer is stabilized by the interaction

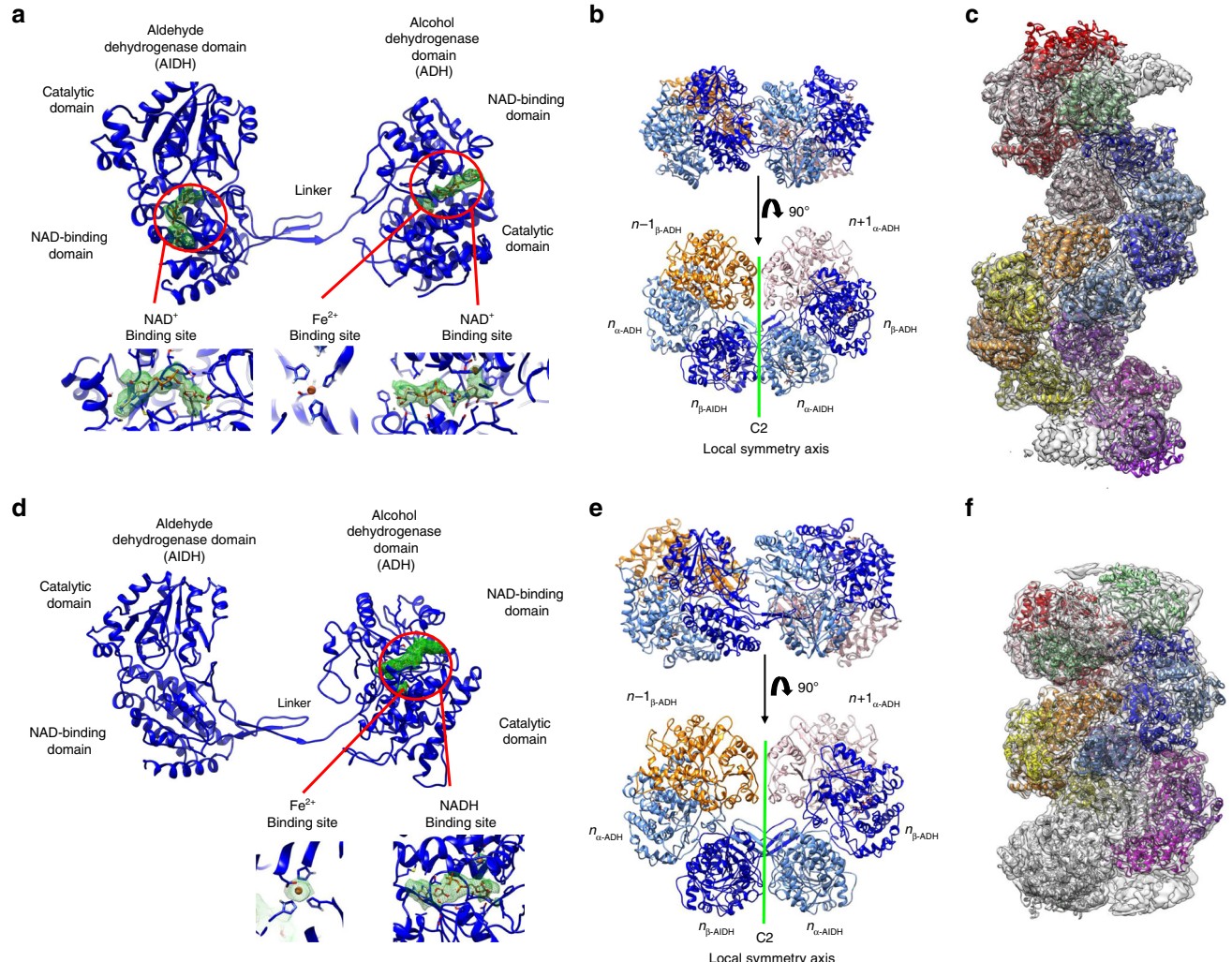

**Fig. 2 Structure of AdhE in extended and compact filaments. a** Structure of the AdhE monomer in the presence of $NAD^+$ and $Fe^{2+}$. The AlDH and ADH domains are identified. In each domain, the electron density corresponding to the $NAD^+$ and $Fe^{2+}$ is displayed in green mesh, as it is visible in the cryoEM map. In the lower part of the panel, larger views of the cofactor-binding sites are provided. **b** Structure of the AdhE dimer in its extended conformation. The AdhE$_n$ dimer is made of the α (in light blue) and β (in blue) protomers. Each protomer is composed of ADH and AlDH domains. The AdhE$_n$ dimer is in interaction with the ADH$_{n-1}$ and ADH$_{n+1}$. The AdhE$_α$ and AdhE$_β$ are related by C2 symmetry. **c** Structural model of the AdhE filament in its extended conformation. This model was obtained by docking the structure of the AdhE dimer shown in **b**, in the cryoEM map obtained by HR and by duplicating this dimer along the filament axis using its helical symmetry. **d** Structure of the AdhE monomer in the presence of NADH and $Fe^{2+}$. The AlDH and ADH domains are identified. In the ADH domain, the electron density corresponding to the NADH and $Fe^{2+}$ is displayed in green mesh, as it is visible in the cryoEM map. No density corresponding to the NADH is visible in the AlDH domain active site. In the lower part of the panel, larger views of the ADH cofactor-binding site are provided. **e** Structure of the AdhE dimer in its compact conformation. The AdhE$_n$ dimer is made of the α (in light blue) and β (in blue) protomers. Each protomer is composed of ADH and AlDH domains. The AdhE$_n$ dimer is in interaction with the ADH$_{n-1}$ and ADH$_{n+1}$. The AdhE$_α$ and AdhE$_β$ are related by C2 symmetry. **f** Structural model of the AdhE filament in its compact conformation. This model was obtained by docking the structure of AdhE dimer shown in **d**, in the cryoEM map obtained by HR and by duplicating this dimer along the filament axis using its helical symmetry.

between the oligomerization domain (residues 87–101) and the linker (residues 440–451) of one AlDH with a β-sheet of the other AlDH catalytic domain. The (αADH)$n$/(β-ADH)$n+1$ interface is stabilized by four salt bridges and hydrogen bonds (Supplementary Figs. 5c and 6c).

**Comparison with monofunctional enzymes**. The structure of AdhE was compared with the structure of monofunctional ADH or AlDH enzymes. We selected homologous monofunctional enzymes with active sites of similar topology than AdhE using the COACH server[22,23]. For AdhE AlDH domain, the crystal structure of *Rhodopseudomonas palustris* propionaldehyde dehydrogenase (PDB 5JFL) was identified. For AdhE ADH domain, the crystal structure

of the lactaldehyde : 1,2-propanediol oxidoreductase of *E. coli* (PDB 2BL4) was identified. These monofunctional enzymes were compared with the corresponding AlDH and ADH domains in AdhE. As shown in Fig. 3a, b, the overall canonical fold found in AlDH and ADH enzymes is conserved in AdhE. Interestingly, the monofunctional enzymes are all active as dimers or tetramers. By superimposing these structures with AdhE, we could observe that the interfaces present in dimers of monofunctional enzymes correspond to the ADH–ADH and AlDH–AlDH interfaces of the extended spirosomes (Fig. 3a, b). Of note, in the AdhE filament in its compact state, the conformation of the AlDH domain and the AlDH–AlDH interface are not superimposable with their counterpart in the monofunctional enzymes.

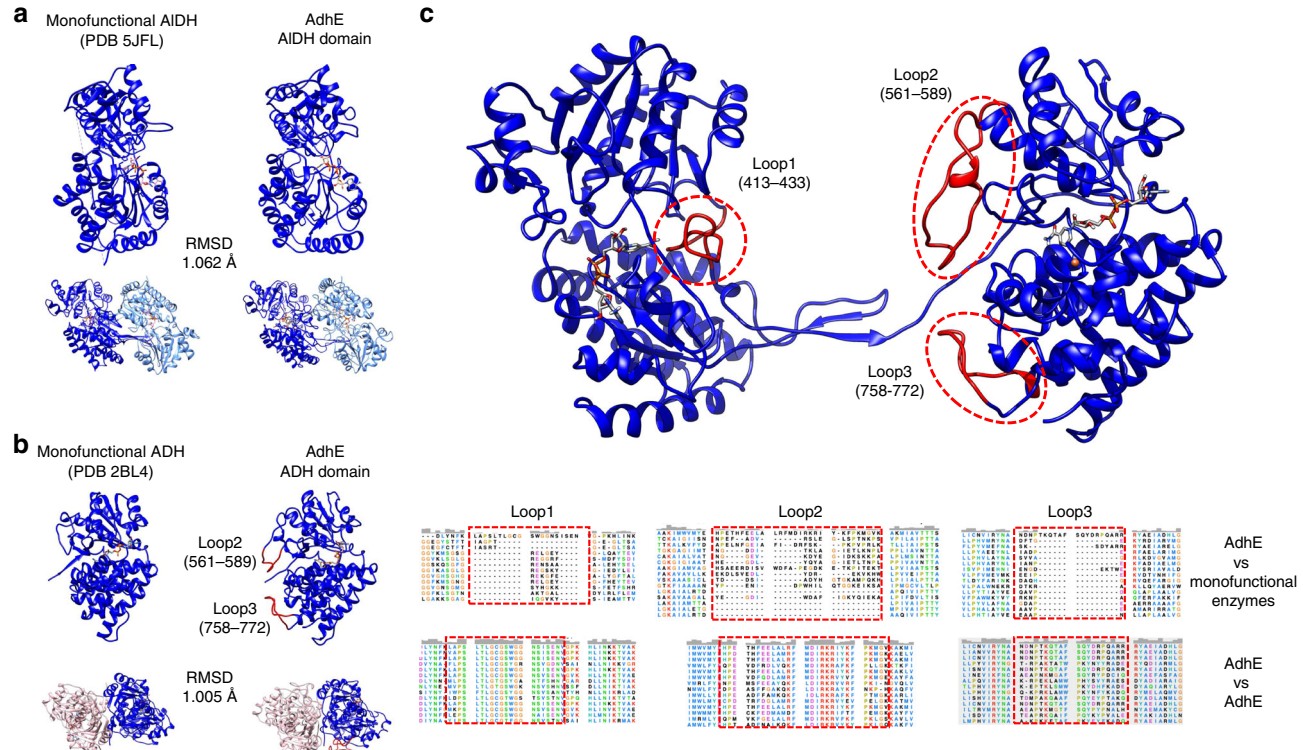

**Fig. 3 Comparison of the structure of AdhE in its extended conformation with monofunctional ADH and AlDH enzymes. a** The structures of the monofunctional enzyme propionaldehyde dehydrogenase from *R. palustris* (left) and AdhE AlDH domain (right) are compared. The monomers share the same overall fold. The root-mean-square deviation of atomic positions (RMSD) between the two structures is indicated in the figure. Both enzymes assemble as dimers, which are superimposable with each other. **b** The structures of the monofunctional enzyme lactaldehyde : 1,2-propanediol oxidoreductase from *E. coli* and AdhE ADH domain are compared. The monomers share the same overall fold. The RMSD between the two structures is indicated in the figure. Both enzymes assemble as dimers, which are superimposable with each other. **c** Localization and conservation of three loops specific to AdhE. These loops are not present in the monofunctional enzymes but are conserved in AdhE homologs.

Sequence alignments between *E. coli* AdhE and monofunctional ADH and AlDH enzymes reveal the presence of three loops that are specific to AdhE (Fig. 3c and Supplementary Data 1 and 2). Importantly, these loops are located close to the AlDH active site (loop1, residues 413–433) or at the AlDH–ADH interface in the AdhE dimer (loops 2 and 3, residues 561–589 and 758–772). Even if their sequence can vary, these loops are always present in the AdhE homologs (Fig. 2c and Supplementary Data 3).

**The interface between the AlDH and ADH domains**. In the monofunctional AlDH and ADH enzymes, access to the active site is possible by a continuous channel embedded between the catalytic and NAD-binding domains (Fig. 4a). On one side, one channel allows the binding of the NAD⁺/NADH (and of Acetyl-CoA for the AlDH) in the catalytic pockets. On the other side of the domains, a channel allows the entrance/exit of the substrates/products of the enzymatic reactions (ethanol and acetaldehyde). As shown in Fig. 4a, these channels are conserved in the AlDH and ADH domains of AdhE. Remarkably, the substrate/product channels of both the AlDH and ADH domains lead to the two cavities located at the AlDH–ADH interfaces within the AdhE dimer. The loops 2 and 3 seal this cavity by mediating the interactions between the AlDH and ADH domains (Fig. 4b). This allows a direct channeling between the AlDH and ADH domain active sites (Fig. 4c).

**Extended spirosomes contain catalytically active AdhE monomers**. The comparison of AdhE structures in the absence of cofactors[15], in the presence of cofactor bound only to the ADH

domain or to both ADH and AlDH domains provides valuable insight about the regulation of AdhE enzymatic activities. Although very little differences are observed between the structures of the Apo-ADH and NADH-bound ADH domains (Supplementary Fig. 4), the binding of NAD⁺ in the AlDH domain induces a significant conformational change in this domain. Cofactor binding induces a closure of the AlDH catalytic site onto the NAD/NADH-binding domain (Fig. 5a). This induced-fit conformational change orients the key catalytic residues (Cys 246, His 367, and Glu 335) in the optimal orientation for catalysis (Fig. 5b). At the level of the whole domain, it modifies the orientation of the catalytic domain relative to the NAD/NADH-binding domain (Fig. 5c). This conformational change induced by the cofactor binding seems to be specific to AdhE. In homologous monofunctional AlDH enzymes, the cofactor binding site and catalytic residues adopt the same topology in the absence and in the presence of cofactor.

**AdhE filamentation is essential for AlDH activity in vitro**. The activities of AlDH and ADH domains within the purified spirosome fractions were recently reported in vitro by using absorbance properties of NADH at 340 nm. We performed similar activity assays either in physiological conditions (Ethanol production) (at pH 7 in the presence of acetyl-CoA and NADH) and in conditions forcing the reverse reaction (from Ethanol to Acetyl-CoA) (pH 8.8 in the presence of ethanol, NAD⁺, and CoA). Our results confirm that both AlDH and ADH domains are functional in vitro within the spirosome fraction (Fig. 6c).

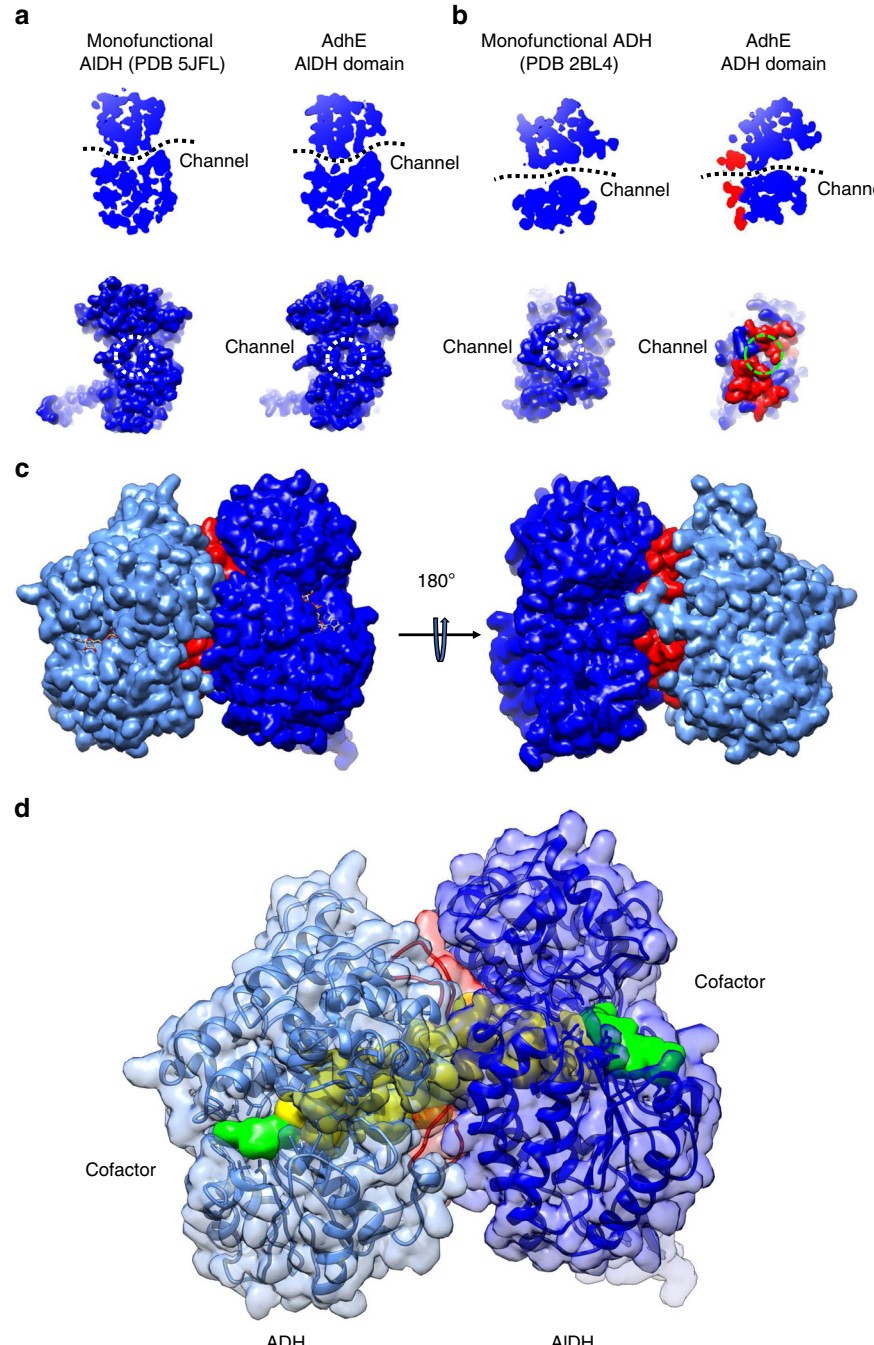

**Fig. 4 A continuous channel between the ADH and AlDH active sites. a** A conserved channel crosses the monofunctional propionaldehyde dehydrogenase from *R. palustris* (left panels). This channel is conserved in the AlDH domain in AdhE (right panel). Top, cut-out slice through the AlDH domain of the bottom, surface representation of the AlDH domain showing the substrate entrance/exit channel (opposite of the NAD-binding channel). **b** A conserved channel crosses the monofunctional lactaldehyde : 1,2-propanediol oxidoreductase from *E. coli* (left panels). This channel is conserved in the ADH domain in AdhE (right panel). Top, cut-out slice through the ADH domain of the bottom, surface representation of the ADH domain showing the substrate entrance/exit channel (opposite of the NAD-binding channel). **c** Surface representation of ADH$_\alpha$ (in light blue) and AlDH$_\beta$ (in blue). The loops 2 and 3 from ADH$_\alpha$ are colored in red. **d** A channel connects directly the active sites of ADH$_\alpha$ (in light blue) and AlDH$_\beta$ (in blue). The domains are represented as transparent surfaces and ribbons with the same color code than above. A surface representation of channel is colored in yellow. Surface representations of NAD$^+$ molecules bound to ADH and AlDH domains are colored in green.

However, the spirosome fraction contains spirosomes but is also contaminated with smaller AdhE assemblies (tetramers, dimers, etc.). In order to confirm that AdhE filaments are active, it is necessary to disrupt AdhE filamentation and monitor AdhE activity. The AlDH–AlDH and ADH–ADH interfaces are conserved across homologous monofunctional AlDH and ADH

enzymes. The vast majority of these monofunctional proteins are active as dimer and the conservation of these interfaces along the filament could be necessary for the activity of the domains. Therefore, we decided to shorten the linker between the AlDH and ADH domain to perturb the dimerization of AdhE. Guided by the structure, we generated a Δ446–449 AdhE

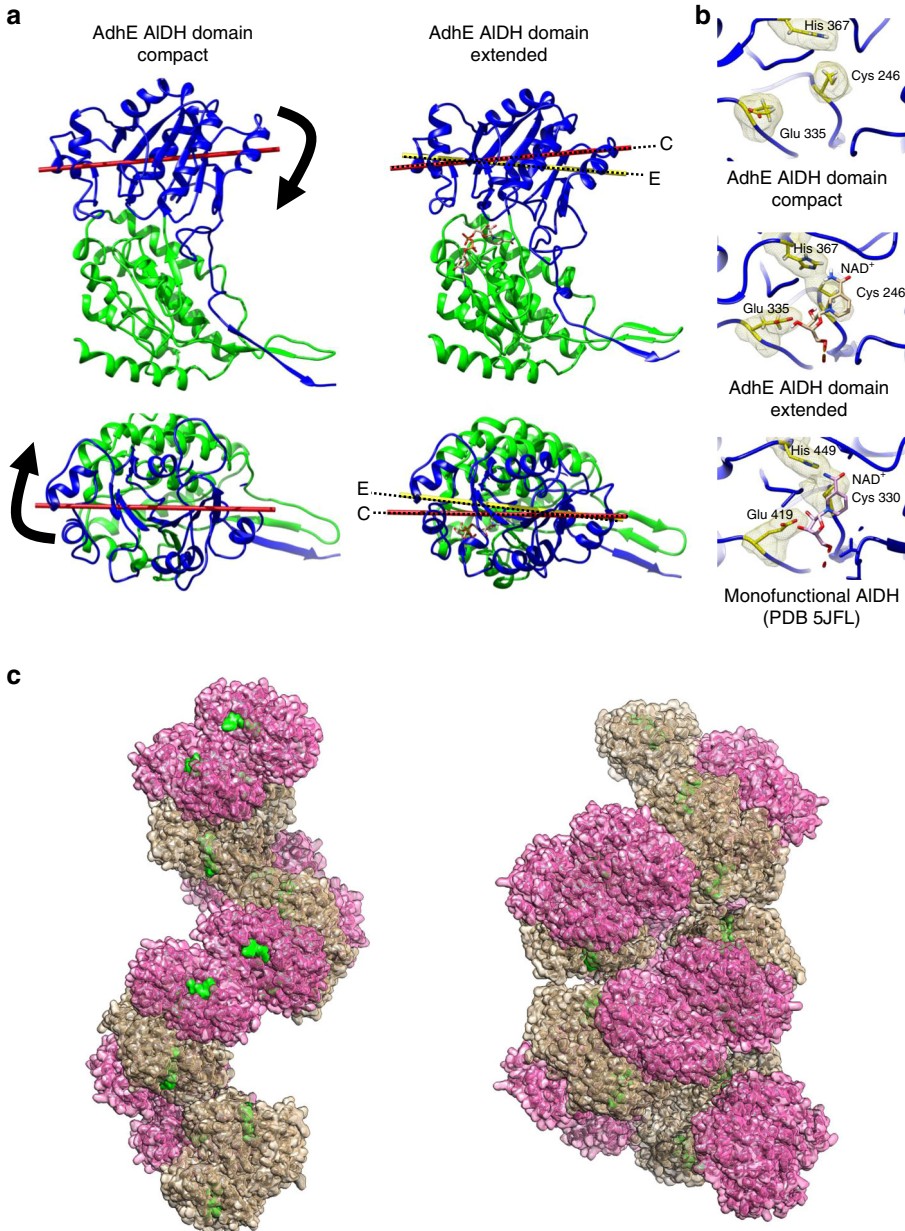

**Fig. 5 Conformational changes between the AdhE compact and extended states. a** Sub-domain motion in the AlDH domain. Ribbon representation of the AlDH domain from AdhE. The NAD-binding domain (residues 1–214) is colored in green. The catalytic domain (residues 215–448) is represented in blue. The NAD-binding domains are represented in the same position for the compact and extended state. The catalytic domain moves relative to the NAD-binding domain between AdhE compact and extended states. The axis of the catalytic domain is represented in red for the compact conformation (C) and yellow for the extended conformation (E). Top, side view of the AlDH domain. Bottom, view from the catalytic domain. **b** Zoomed view of the AlDH catalytic site of compact and extended AdhE and monofunctional propionaldehyde dehydrogenase from *R. palustris*. The proteins are represented as blue ribbons. The catalytic histidine, glutamate and cysteine are represented as sticks and volumes. They are colored in yellow. The NAD$^+$ molecules are represented as sticks. **c** Surface representation of the AdhE filament in its extended (left) and compact (right) conformation. The ADH domains are colored in beige and the AlDH domains are colored in pink. The NAD$^+$ molecules are colored in green.

mutant. We showed that this deletion prevents the filamentation of AdhE (Fig. 6b and Supplementary Fig. 7a). Negative stain EM shows that the Δ446–449AdhE mutant does not assemble into long filament as Wild-Type AdHE (WT AdhE) (Fig. 6b). Size-exclusion chromatography experiments coupled to multi-angle laser light scattering (SEC-MALS) experiments determined that this mutant has a molecular mass of 200 kDa, suggesting that a dimer still formed (Supplementary Fig. 7b). The activities of the AlDH and ADH domains of Δ446–449 AdhE were also monitored. In physiological condition (pH 7), the Δ446–449 AdhE activity was drastically reduced compared with the WT AdhE activity (Fig. 6c). Indeed, the specific activities calculated for the WT AdhE and Δ446–449 AdhE were 0.112 and 0.035 mmol min$^{-1}$ mg$^{-1}$, respectively. In reverse condition, the Δ446–449 AdhE activity (0.351 mmol min$^{-1}$ mg$^{-1}$) was not affected compared with WT AdhE (0.336 mmol min$^{-1}$ mg$^{-1}$). These results show that the activity of the AlDH domain was strongly affected, while the ADH domain was still functional when AdhE filamentaion is disrupted.

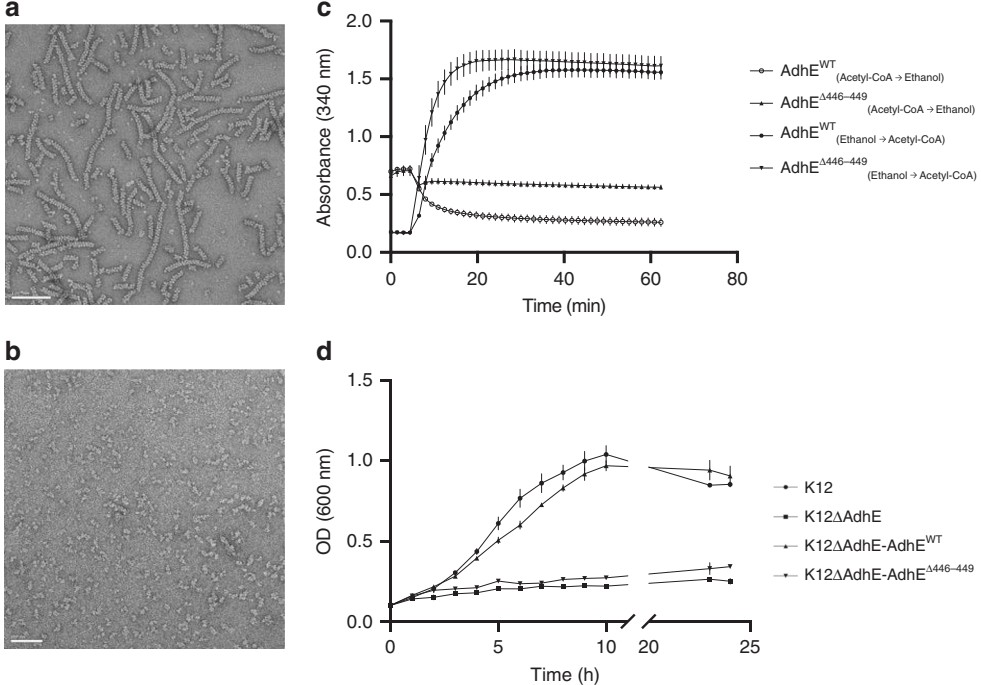

**Fig. 6 Functional analysis of AdhE in vitro and in vivo. a** Negative stain electron micrograph of purified Wild-Type AdhE filaments. Scale bar 100 nm. **b** Negative stain electron micrograph of purified Δ446–449 AdhE filaments. Scale bar 50 nm. **c** Enzymatic assays monitoring AdhE activities of Wild-Type and Δ446–449 mutant AdhE. The conditions of each assay are indicated in the legend of the graph. The errors bars show SEM (3). **d** Growth curves for K12 *E. coli* strains in fermentative conditions in minimal medium. The strains used are indicated in the legend of the graph. The errors bars show SEM (3).

**Filamentation is essential for AdhE activity in vivo**. The importance of AdhE filamentation on AdhE activity was established in vitro (this work and ref. [15]). However, it was still unclear whether it was essential for AdhE activity in vivo during fermentation. To test the activity of the full-length AdhE and the Δ446–449 AdhE in vivo, complementation assays were carried out using a ΔAdhE K12 strain[24] as described in Methods section. ΔAdhE *E. coli* was transformed with plasmid PKG116 expressing full-length AdhE, leading to restoration of bacterial growth in minimal medium in a low concentration of oxygen. In contrast, the strain transformed with PKG116 expressing Δ446–449 AdhE was not able to grow under the same conditions (Fig. 6d). These results show that the filamentation of AdhE and functional AlDH enzymatic activity is not only necessary for ethanol production during fermentation but also for regenerating NAD$^+$ pool required for bacterial growth. The presence of the full-length AdhE and the Δ446–449 AdhE was confirmed by western blotting using anti-AdhE antibody (Supplementary Fig. 8a, b).

## Discussion

Recently, a study reported the structure of the compact conformation of the AdhE spirosomes[15] in the absence of any cofactor bound. Here we present the structure of the spirosomes in the presence of cofactors in their compact and extended conformation. We reveal the specific organization between AdhE monomers with three different interfaces along the filament as follows: AlDH–AlDH interface, ADH–ADH interface, and AlDH–ADH interface. Although the AlDH–AlDH and ADH–ADH interfaces are conserved in the corresponding monofunctional enzymes, the AlDH–ADH interface is specific to AdhE homologs. At this interface, specific loops inserted in the ADH domain (loops 2 and 3) define a channel connecting directly the active sites of one ADH domain to its adjacent AlDH domain. During alcoholic fermentation, this substrate channeling

prevents the acetaldehyde produced by the AlDH domain, a highly toxic intermediate, to be released in the cytoplasm. In addition, this channeling is probably important for an efficient coupling of the AlDH and ADH enzymatic activities. Interestingly, bacteria evolved other strategies to confine multi-enzymatic reactions and prevent the release of toxic intermediates[25]. In *Salmonella*, bacterial microcompartment resembling viral capsids encapsulate aldehyde and ADH activities to optimize catalysis and prevent the release of toxic intermediates[26].

Early studies showed that the addition of cofactors triggers a conformational change in *E. coli* spirosomes. Although these results were recently confirmed, the molecular basis of this phenomenon was not understood. In our study, we show that cofactor binding to both the ADH and AlDH domains of AdhE is required to trigger AdhE filament extension. Therefore, only extended filaments are bound to cofactors and are able to catalyze the two-step reaction of conversion of acetyl-CoA to ethanol. Our structural data show that, in the compact AdhE filament, the AlDH domain is in a conformation that is not favorable for cofactor binding and for catalysis. We show that cofactor binding to this domain triggers an induced-fit conformational change, which activates this active site and induces the closure motion of the AlDH$_n$ catalytic domain onto the cofactor binding domain in the AlDH domain. In turn, this motion is transmitted to the ADH$_n$–ADH$_{n+1}$ dimer, which behaves as a rigid body moving towards the center and extremity of the filament. As they allow a tight contact between the AlDH and ADH domains, the ADH loops 2 and 3 are essential to the transmission of this motion. The AlDH$_{n+1}$ domain contacting the ADH$_{n+1}$ is in turn affected by the motion and switches its active site into an active conformation without the need of cofactor binding. We propose that, in the extended filament, a fraction of the AlDH domains bound to cofactors maintains the other AlDH active sites in an active conformation, optimal for substrate and cofactor binding and for catalysis. The observation that AlDH activity is impaired when

AdhE filamentation is prevented supports the hypothesis of a cooperative behavior between AlDH active sites within the AdhE filament. In the mutants generated by us and others, the AlDH active site or dimerization interface is not directly affected by the mutation(s). However, each AlDH active site of the AdhE dimer needs to convert from inactive to active conformation at each catalytic cycle. Interestingly, we do not observe any filament extension in the presence of NADH. While this cofactor is bound to the ADH domain, its binding to the AlDH domain could be dependent on the covalent binding of the acetyl moiety to the catalytic cysteine residue or on the binding of NAD+ to other AlDH domains within the filament. Furthermore, it has already been reported for the betaine aldehyde dehydrogenase[27] that the kinetic of NADH binding could be affected by the prior binding of NAD+ or the aldehyde.

More than 20 enzymes self-assemble as polymers (as reviewed in ref. [28]). For the vast majority of them, polymer formation is essential for enzyme activity. For AdhE, we confirm the previous observation that AdhE filamentation is necessary for its activity in vitro. In addition, we show that this filamentation is essential in vivo during fermentation and anaerobic growth in minimal medium. However, until now, the exact role of filamentation in AdhE function was unclear. Indeed, the recent structure of the spirosome in their compact conformational state[15] did not bring any clear answer to this question.

We propose that the conformational switch from compact to extended conformation could be used by the bacteria to regulate AdhE activity. The compact conformation would be a storage form of inactive AdhE protomers. In low oxygen conditions, the pool of NADH to be oxidized to NAD+ to be used in the glycolytic process increases in the bacterial cell. NADH binding to some AdhE subunits within the spirosomes would have a cooperative effect to convert the whole spirosome from compact to extended conformation and quickly activate all the AdhE subunit within the whole filament. The acetyl-CoA binding would be required for binding of the NADH to the AlDH domain. As long as the pool of NADH stays high within the cell, the AdhE filaments would stay extended and highly active. Interestingly, native spirosomes isolated from other bacteria or unicellular eukaryotic organisms all display an extended conformation. For example, we previously showed that spirosomes isolated from *Clostridium difficile* or streptococcal species, all grown in fermentative conditions, are in an extended conformation. It remains to establish whether these filaments are intrinsically extended or share the same conformational switch with *E. coli* spirosomes. Nevertheless, molecules that would prevent or stimulate the extension of AdhE filaments would be valuable tools to impair fermentation in pathogens or optimize the alcoholic fermentation process in biotechnology.

## Methods

**Spirosome expression and purification**. The gene encoding AdhE from *E. coli* (AdhE*E.coli*) was amplified from *E. coli* K12 genomic DNA (Supplementary Fig. 9) and cloned into the pET15 expression vector resulting in a sequence coding for a N-terminal hexahistidine tagged AdhE (His6-AdhE). Plasmids from a single clone were isolated, checked by sequencing and transformed into *E. coli* BL21(DE3) bacteria. Bacteria were grown in Luria-Bertani (LB) media at 37 °C, 180 r.p.m. to exponential phase. From $OD_{600} = 0.6$–0.7, overexpression was induced by adding 1 mM isopropryl β-D-1 thiogalactopyranoside for 16 h at 16 °C. Then, cells were pelleted at $6000 \times g$ for 30 min and resuspended in lysis buffer (50 mM Tris-HCl pH 7.5, 200 mM NaCl, cOmplete Protease Inhibitor Cocktail, DNase (1.25 µg ml$^{-1}$), lysozyme (100 µg ml$^{-1}$)). Cells were sonicated and debris were pelleted at $11,000 \times g$ for 1 h. The supernatant was loaded on a nickel-NTA affinity column (GE Healthcare) and His6-AdhE was eluted with 300 mM imidazole. Fractions containing AdhE were loaded on a Superdex200 (GE Healthcare) size-exclusion column. Purity of AdhE was confirmed with 12% SDS-polyacrylamide gel electrophoresis. The presence of spirosomes was checked by negative stain electron microscopy. Five microliters of sample was applied on a glow-discharged carbon-coated grids (300 mesh Cupper, EMS). After two washes with water and

one wash with a drop of 0.75% uranyl formate, the sample was stained for 1 min with one drop of uranyl formate and blotted with Whatman paper. Micrographs were collected using CM120 microscope (FEI) operated at 120 kV equipped with charge-coupled device camera.

**Cryo-electron microscopy**. Freshly purified protein was incubated with 5 mM NAD+, 5 µM CoA, 0.3 mM Fe(NH4)(SO4)2, and 3 mM MgSO4, and applied to cryoEM grids. Samples were vitrified with a Vitrobot (ThermoFisher) at 4 °C at 100% humidity. Four microliters of sample was applied onto glow-discharged (Elmo Cordouan) grids. The excess of sample was immediately blotted away (4 s) with Whatman paper and plunged into liquid ethane cooled by liquid nitrogen. Movies were recorded on Talos Arctica (ThermoFischer) operated at 200 kV equipped with K2 direct electron detectors at ×36,000 of magnification and a pixel size of 1.13 Å per pixel using SerialEM[29]. Micrographs were collected in a defocus range of −0.4 µm to −2 µm and with a dose of 0.77 electrons per Å$^2$ per frame.

**Image processing and 3D reconstruction**. Movies were aligned for beam-induced motion using MotionCor2 and CTF parameters were assessed using GCTF. The following steps were performed using RELION 3.0 software. Details and statistics about each dataset are provided in the Table 1.

For SPA approach, non-overlapping segments of the AdhE filaments were manually picked and particles were extracted using a box size of 260 pixels. These particles were 2D classified in ten classes. The 2D classes corresponding to distinct orientations of the filaments were selected and used as references to automatically pick segments in all the micrographs After extraction, several rounds of 2D classification were performed. From a subset containing 10,000 particles, an initial 3D map was reconstructed without imposing symmetry. The whole dataset was then used to refine this 3D map. A homology model of *E. coli* AdhE was generated with SWISS-MODEL[19] using the crystal structure of *Vibrio parahaemolyticus* acetaldehyde dehydrogenase (PDB code 3MY7) of the *V. parahaemolyticus* AdhE and the structure of *Geobacillus thermoglucosidasius* ADH domain (PDB code 3ZDR)[16] of *G. thermoglucosidasius* AdhE. The generated AdhE model was fitted in the cryoEM map using UCSF Chimera[30]. A 3D mask was generated in Chimera to contain only one AdhE dimer and the adjacent ADH domains. This 3D mask was then used for focused refinement in RELION. After CTF refinement and Bayesian polishing, the 3D map was further refined by focused refinement. The densities outside of the 3D mask used above for focused refinement were subtracted and a final refinement was performed. For the map of the extended AdhE filament, C2 local symmetry was applied during this refinement to obtain the final map. The final resolution was calculated with two masked half-maps, using 0.143 Fourier shell correlation (FSC) cut-off criterion. Local resolution was estimated using RELION (Supplementary Fig. 3). Map visualizations were prepared using UCSF Chimera.

For HR, the procedure was similar with the following differences. During manual and automatic picking, overlapping segments of the filaments were used. No initial model was generated. Instead, a cylinder of 150 Å in diameter was used. A first refinement without any symmetry imposed was performed. The helical twist and rise were determined in real space on this initial model using relion_helix_toolbox (Rotational search and correlation along Z) Then, these helical parameters were used and refined in further refinements.

**Model building and refinement of atomic models**. For the model of AdhE in its extended state, a homology model of AdhE*E.coli* was fitted in the refined SPA cryoEM map with C2 local symmetry. Using these models as starting points, an initial 3D model of the AdhE dimer and adjacent ADH domains was manually built in Coot[31]. The map was sharpened in PHENIX[20,21] (phenix.autosharpen). The final model was refined by several rounds of manual refinement in Coot software and real-space refinement using phenix.real_space_refine with non-crystallographic symmetry (NCS) restrains. The model was validated using MolProbity[32] and phenix.validation_cryoem implemented in PHENIX software.

For the model of AdhE in its compact state, the models of the AlDH and ADH domains obtained for the extended state were docked in the cryoEM map obtained by SPA. The cryoEM map was sharpened in PHENIX (phenix.autosharpen). An initial 3D model of the AdhE dimer and adjacent ADH domains was manually built in Coot. The final model was refined by several rounds of manual refinement in Coot software and real-space refinement using phenix.real_space_refine with NCS and reference model restrains. The reference model used was the model obtained for AdhE in its extended state. The model was validated using MolProbity and phenix.validation_cryoem implemented in PHENIX software.

For the models of the AdhE filaments in their extended and compact states, the corresponding model of the AdhE dimer were docked into the cryoEM HR maps obtained. These maps were sharpened in PHENIX and symmetrized in RELION using the refined helical parameters. The model was refined using using phenix.real_space_refine with NCS and reference model restrains. The reference model used was the AdhE model in its extended state. The model was validated using MolProbity and phenix.validation_cryoem implemented in PHENIX software.

**Table 1 Data collection and processing.**

| | Extended ($Fe^{2+}$–$NAD^+$–CoA) | | Compact ($Fe^{2+}$–NADH) | |
|---|---|---|---|---|
| Data collection | | | | |
| Microscope–camera | Talos Arctica–K2 summit | | Talos Arctica–K2 summit | |
| Voltage (kV) | 200 | | 200 | |
| Magnification | 36,000 | | 36,000 | |
| Electron exposure ($e^-$ per $Å^2$) | 0.77 | | 0.77 | |
| Pixel size (Å) | 1.13 | | 1.13 | |
| Decofus range (um) | 0.4 – 2.4 | | 0.4 – 2.4 | |
| Processing | Helical reconstruction | Single-particle analysis | Helical reconstruction | Single-particle analysis |
| Symmetry imposed | C1 | C1 | C1 | C1 |
| | Rise 56.5 Å, Twist 164.4° | C2 | Rise 34.7 Å, Twist 154.5° | |
| Initial particle images (no.) | 821,557 | 1,236,367 | 227,537 | 599,988 |
| Final particle images (no.) | 203,288 | 138,927 | 98,677 | 226,646 |
| Map resolution (Å)–(FSC threshold model) | 3.8-(0.143) | 3.4-(0.143) | 5-(0.143) | 3.9-(0.143) |
| Refinement and validation | | | | |
| Map sharpening (B-factor) ($Å^{-2}$) | 121.41 | 82.58 | 246.65 | 206.89 |
| Model composition | | | | |
| No. of chains | | 6 | | 6 |
| Atoms (no.) | | 39,781 | | 39,760 |
| Residues (no.) | | 2576 | | 2578 |
| Ligands (no.) | | $Fe^{2+}$ (2)-$NAD^+$ (4) | | $Fe^{2+}$ (2)-NADH (2) |
| Bond lengths (Å) | | 0.011 | | 0.006 |
| Bond angles (°) | | 0.989 | | 1.587 |
| Ramachandran favored % | | 87.46 | | 85.02 |
| Ramachandran allowed % | | 12.54 | | 14.05 |
| Ramachandran outliers % | | 0.00 | | 0.93 |
| Rotamers outliers % | | 1.45 | | 3.67 |
| MolProbity score | | 2.11 | | 2.59 |
| Clashscore | | 6.89 | | 9.43 |
| CC (mask) | | 0.82 | | 0.78 |
| CC (box) | | 0.83 | | 0.77 |
| CC (peaks) | | 0.73 | | 0.60 |
| CC (volume) | | 0.82 | | 0.79 |
| Mean CC for ligands | | 0.76 | | 0.71 |

**Mutagenesis**. $His_6$-AdhE-pET15 plasmid was amplified using 5′-AACATGTTGT GGGCACAAAC-3′ and 5′-AGCAACGGTTTTCTTGTTG-3′, to delete the coding region corresponding of 446–449 amino acids. After DpnI digestion, amplified fragment was phosphorylated using T4 polynucleotide kinase (NEB) and then ligated using T4 DNA ligase (NEB). DH5α were transformed with this plasmid. Plasmids from one single clone were extracted and checked by sequencing.

**Multi-angle laser light scattering**. SEC-MALS and refractometry were performed on a Superdex S200 5/150 GL increase column (GE Healthcare). Twenty-five microliters of AdhE mutant protein were injected at a concentration of 10 mg $ml^{-1}$ in buffer 0.05 M Tris-HCl pH 7.5, 0.2 M NaCl. On-line MALS detection was performed with a miniDAWN-TREOS detector (Wyatt Technology Corp., Santa Barbara, CA) using a laser emitting at 690 nm and by refractive index measurement using an Optilab T-rex system (Wyatt Technology Corp., Santa Barbara, CA). Weight averaged molar masses were calculated using the ASTRA software (Wyatt Technology Corp., Santa Barbara, CA).

**Enzymatic assays**. Enzymatic assays were performed at 37 °C by following the absorbance of NADH at 340 nm with infinite M1000 plate-reader (TECAN). Reactions were started by adding of 0.04–4 µM of purified enzyme. Reductase activities of AlDH–ADH were monitored using 0.2 mM acetyl-CoA and 0.4 mM NADH in Tris-NaCl buffer pH 7.5. Dehydrogenase activities of AlDH-ADH were monitored using 1% ethanol, 1.5 mM $NAD^+$, and 0.2 mM CoA in Tris-NaCl buffer pH 8.8. Three independent experiments were performed each in triplicate.

**Anaerobic growth of complemented Δadhe K12 strain**. The Δadhe K12 strain was provided by the Keio Collection[24]. Two versions of the low copy PKG116 plasmid were constructed using AQUAcloning method: PKG116-AdhE^WT and PKG116-AdhE^Δ446–449. Each plasmid was incorporated into the Δadhe K12 strain using electroporation technique. Bacteria were grown in LB medium containing 1 µM of sodium salicylate and then back diluted at $OD_{600}$ = 0.1 in minimum media. Anaerobic growth curves were monitored using optical density ($OD_{600}$).

Three independent experiments were performed each in triplicate using K12 WT as a positive control. The expression of AdhE was checked by western blot analysis using anti-AdhE antibody (AS10748, Agrisera) diluted 1/1000 coupled with a secondary anti-rabbit-HRP (AB_10015282, Jackson ImmunoResearch) antibody diluted 1/5000.

**Reporting summary**. Further information on research design is available in the Nature Research Reporting Summary linked to this article.

## Data availability

The cryoEM maps of *E. coli* spirosomes have been deposited in the Electron Microscopy Data Bank under ID codes EMD-10551 and EMD-10552, for the extended spirosomes obtained by SPA and HR, respectively, and under ID codes EMD-10555 and EMD-10631 for the compact spirosomes obtained by SPA and HR, respectively. The atomic coordinated for AdhE in its extended and compact conformation have been deposited in the PDB under ID codes PDB 6TQH and 6TQM, respectively. The source data underlying Fig. 6c, d and Supplementary Figs. 1a and 11a, b are provided as a Source Data file. Other data are available from the corresponding authors upon reasonable request.

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

## Acknowledgements

This work was supported by University of Bordeaux, CNRS, and ERC CoG TransfoP-neumo. P.P. holds a PhD fellowship from University of Bordeaux. We thank Claire Stines-Chaumeil for advice in Enzymology and Armel Bezault from the cryoEM facility at IECB.

## Author contributions

P.P. performed all the experiments except the MALLS experiments performed by L.T. P.P. helped by C.R. and R.F. solved the cryoEM structures and analyzed them. R.F. and E.M. designed and supervised the work. R.F. and P.P. wrote the manuscript.

## Competing interests

The authors declare no competing interests.
