## [Peer Review File · Nature Communications]

Reviewers' comments:

Reviewer #1 (Remarks to the Author):

Pony et al have addressed the structure of the filamentous AdhE from *Escherichia coli* and the effect of filamentation on activity. Unfortunately, in the process of this study a work from Kim et al was published in the same venue that described the filamentous structure of this enzyme and the role of filamentation in activity. Therefore, the manuscript by Pony et al is largely confirmatory and this is actually stated by the authors at several places in the manuscript. I do not agree with their claim that their study goes far beyond the published study by Kim et al because they have addressed for the first time the physiological role of filamentation. Therefore, they have produced a non-filamentous variant of AdhE in *E. coli* that could not restore growth of an AdhE mutant under anoxic conditions. However, the Western Blot presented in sFig. 10 is not convincing and I am not convinced that that protein is actually produced. Anyway, the manuscript is largely confirmatory and not suitable for publication in a high-impact journal.

Reviewer #2 (Remarks to the Author):

This article looks at the structure and function of filamentous acetaldehyde-alcohol dehydrogenase enzymes in both a compact and extended form. The authors determined the structures by cryo-electron microscopy and built atomic coordinates into the resulting densities. Based on activity assays, mutagenesis studies, and a careful investigation of their structures, the authors propose a model for the function of the filaments, and the mechanism for expansion into an active form.

This study is well-written, well thought out and concise. It addresses key aspects of the biology, makes solid conclusions from the experiments, and proposes a realistic and plausible model for the mechanism of action. I believe the structures and data presented here are novel, of interest to the readership, and warrant publication in *Nature Communications*. There is a growing body of evidence that oligomerisation (filamentous or otherwise) is an important mechanism of regulation, and this constitutes a clear and compelling example of this.

There are a few points that should be addressed for the sake of clarity.

Comments:

Line numbers missing

Required changes:

Fig4 - Nicely done!!, but the figure legend is unclear. You refer to left panels, but seem to mean 1st and 3rd (is that correct?). Consider rephrasing. (nice images, though!)....

On closer inspection, it seems there is something altogether wrong with the organisation of the panels in Fig4. My version has no 4d!!

Minor changes:

page5 - "therefore, the extended...could be functionally active form ..." why? This does not follow from the previous statements, though it may be true in light of your findings.

Page8 - The dimer is maintained through.... refer to a figure here to aid the reader.

Page12 - "In reverse condition..." - I am not clear on what this means. Please rephrase.

Fig2b - what is the relationship between top and bottom images? 90 rotation? show which axis. If it's supposed to show the interaction with the neighbouring protomers, then they aren't close enough to be in contact. This panel is confusing.

Fig2b and e have the same heading in the legend, but are different structures. be more specific:

extended, compact, ligand, etc.

Fig3a,b - you state the structures are superimposable, but that just means they overlay, and says nothing about the quality of the fit. Give an RMSD, or color by alpha carbon distance.

Page 23: what acquisition software did you use? Citation?

Page24: Helical parameters were determined in real space... How? Describe the method. Shifting, rotating and fitting in Chimera? Rotational search and correlation along Z?

Minor Grammatical changes:

Page3 - In "the" absence of ... (2x)

page4 - 1970s (it's already the 20's again already)

Page5 - The adhE... this belongs in the materials and methods. This section can be shortened, or largely eliminated.

Page6 - does not impair "the"...

Page6 - ... conditions were used in "terms" of...

Page7 - "of the maps (30%) "... should maybe read something like "... symmetry parameters to the central 30% of the maps"

Page11 - in "the" absence...

page11 - ...fractions "were" recently...

page12 - are active as "dimers" and...

page12 - ...along the filament "could be necessary"

page12 - ...interfaces "are" conserved across...

page 12 - "SEC-MALS"

page 13 - in "a" low concentration of oxygen

page 13 - in "the" presence/absence x2

Page 14 - "AIDH_n" should probably have the "n" (and later n+1) as subscript for clarity

Page24 - after "CTF" refinement ... was further "refined"

Page24 - used above for "focused"

Mike Strauss

Itemized response to Reviewers' comments:

Reviewer #1 (Remarks to the Author):

Pony et al have addressed the structure of the filamentous AdhE from *Escherichia coli* and the effect of filamentation on activity. Unfortunately, in the process of this study a work from Kim et al was published in the same venue that described the filamentous structure of this enzyme and the role of filamentation in activity. Therefore, the manuscript by Pony et al is largely confirmatory and this is actually stated by the authors at several places in the manuscript. I do not agree with their claim that their study goes far beyond the published study by Kim et al because they have addressed for the first time the physiological role of filamentation. Therefore, they have produced a non-filamentous variant of AdhE in *E. coli* that could not restore growth of an AdhE mutant under anoxic conditions. However, the Western Blot presented in sFig. 10 is not convincing and I am not convinced that that protein is actually produced. Anyway, the manuscript is largely confirmatory and not suitable for publication in a high-impact journal.

While we agree with Reviewer 1 that our study confirms the structure of the spiroosomes in their compact conformational state published by Kim et al. in 2019, we respectfully disagree that the other results presented in our manuscript are not suitable for publication in *Nature Communications*. Our results provide important and novel information concerning the structure and function of the spiroosomes. Many key features of these filaments were missed and key questions about the function of these enzymes sometimes eluded in the study published by Kim et al. By determining the structure of the spiroosomes in both their extended and compact states, we had access to many new structural and functional features. We could show that the spiroosomes are only active in their extended conformation. We reveal the channelling mechanism between the ADH and AIDH domains within the filament. On the functional level, Kim et al. Revealed that the AdhE is active within the filaments. By disrupting the ADH-ADH interface, they show that the spiroosomes do not form and that AdhE is no longer active in vitro. By using a mutant of the linker between ADH and AIDH domains that AdhE filamentation we confirm is essential for activity in vitro. While Kim et al. could not explain how AdhE filamentation is linked to its activity, we show that filamentation is required to stabilize AdhE in its active extended conformation. Indeed, within the extended filaments, co-factor binding in AIDH active site will trigger and stabilize long-range activation of the other AIDH active sites. Based on these data, we propose an interesting mechanism for the regulation of the activity of the enzyme. Finally, we show that AdhE filamentation is essential for bacterial growth in fermentative conditions.

We agree with Reviewer 1 that the Western blot provided in sFig. 10 is not 100% convincing. We reproduced the experiment and loaded more material on the SDS-PAGE. An updated Western blot is now provided in supplementary Fig 11. It shows that Wild-type (WT) and mutant AdhE are expressed at the same level in the complemented strains. While the plasmid carrying WT adhE restores fermentation, mutant adhE does not restore growth in micro-aerobic conditions, showing that AdhE filamentation is also essential for bacterial growth in fermentative conditions.

Reviewer #2 (Remarks to the Author):

This article looks at the structure and function of filamentous acetaldehyde-alcohol dehydrogenase enzymes in both a compact and extended form. The authors determined the structures by cryo-electron microscopy and built atomic coordinates into the resulting densities. Based on activity assays,

mutagenesis studies, and a careful investigation of their structures, the authors propose a model for the function of the filaments, and the mechanism for expansion into an active form.

This study is well-written, well thought out and concise. It addresses key aspects of the biology, makes solid conclusions from the experiments, and proposes a realistic and plausible model for the mechanism of action. I believe the structures and data presented here are novel, of interest to the readership, and warrant publication in Nature Communications. There is a growing body of evidence that oligomerisation (filamentous or otherwise) is an important mechanism of regulation, and this constitutes a clear and compelling example of this.

We thank the Reviewer 2 for his assessment of our work.

There are a few points that should be addressed for the sake of clarity.

Comments:

Line numbers missing

Required changes:

Fig4 - Nicely done!!, but the figure legend is unclear. You refer to left panels, but seem to mean 1st and 3rd (is that correct?). Consider rephrasing. (nice images, though!)....

On closer inspection, it seems there is something altogether wrong with the organisation of the panels in Fig4. My version has no 4d!!

The figure 4 was corrected accordingly.

Minor changes:

page5 - "therefore, the extended...could be functionally active form ..." why? This does not follow from the previous statements, though it may be true in light of your findings.

We agree. We removed the sentence "the extended form of the AdhE filaments could be the functionally active form of the filament". This could not be inferred from previous studies.

Page8 - The dimer is maintained through.... refer to a figure here to aid the reader.

Reference to figure 2 and Supplementary Figure 5 was added.

Page12 - "In reverse condition..." - I am not clear on what this means. Please rephrase.

We rephrased this sentence to "We performed similar activity assays either in physiological conditions (Ethanol production) (at pH 7 in presence of acetyl-CoA and NADH) and in conditions forcing the reverse reaction (from Ethanol to AcetylCoA) (pH 8.8 in presence of ethanol, NAD⁺ and CoA).

Fig2b - what is the relationship between top and bottom images? 90 rotation? show which axis. If it's supposed to show the interaction with the neighbouring protomers, then they aren't close enough to be in contact. This panel is confusing.

Fig2b and e have the same heading in the legend, but are different structures. be more specific: extended, compact, ligand, etc.

We corrected the Figure 2b and its legend accordingly.

Fig3a,b - you state the structures are superimposable, but that just means they overlay, and says nothing about the quality of the fit. Give an RMSD, or color by alpha carbon distance.

We indicated the RMSD between the structures in the figure 3a and b. and mentioned in the figure legend.

Page 23: what acquisition software did you use? Citation?

Corrected. The information is now provided in the methods section

Page24: Helical parameters were determined in real space... How? Describe the method. Shifting, rotating and fitting in Chimera? Rotational search and correlation along Z?

Corrected. The information is now provided in the methods section

Minor Grammatical changes:

Page3 - In "the" absence of ... (2x)

Corrected

page4 - 1970s (it's already the 20's again already)

Corrected

Page5 - The adhE... this belongs in the materials and methods. This section can be shortened, or largely eliminated.

We believe this section is necessary for the reader to understand how the spiroosomes were purified.

Page6 - does not impair "the"...

Corrected

Page6 - ... conditions were used in "terms" of...

Corrected

Page7 - "of the maps (30%) "... should maybe read something like "... symmetry parameters to the central 30% of the maps"

Corrected

Page11 - in "the" absence...

Corrected

page11 - ...fractions "were" recently...

Corrected

page12 - are active as "dimers" and...

Corrected

page12 - ...along the filament "could be necessary"

Corrected

page12 - ...interfaces "are" conserved across...

Corrected

page 12 – "SEC-MALS"

Corrected

page 13 - in "a" low concentration of oxygen

Corrected

page 13 - in "the" presence/absence x2

Corrected

Page 14 - "AIDH_n" should probably have the "n" (and later n+1) as subscript for clarity

Corrected

Page24 - after "CTF" refinement ... was further "refined"

Corrected

Page24 - used above for "focused"

Mike Strauss